# Automata Learning for Neural Event ODEs: An Interpretable Model of Piecewise Dynamics

## Abstract

Discrete events within a continuous system cause discontinuities in its derivatives. Given event specifications and state update functions, ODE solvers can integrate until an event, apply the update function, and restart the integration process to obtain a piecewise solution for the system. However, in many real-world scenarios, the event specifications are not readily available or vary across different black-box implementations. We present a method to learn the dynamics of a black-box ODE implementation that uses abstract automata learning and Neural Event ODEs. Without prior knowledge of the system, the method extracts the event specifications and state update functions and generates a high-coverage training dataset through abstract automata learning. We learn interpretable models of a bouncing ball's Zeno behavior, the symmetry of heating and cooling processes with a thermostat-controlled water heater, and a switching dynamical system without prior knowledge of their underlying ODEs. Additionally, our approach introduces an efficient training process for Neural Event ODEs that slices training trajectories into temporally consecutive pairs within continuous dynamics. Both contributions ensure well-posed initial values for each ODE slice. A proof-of-concept implementation captures event specifications in an interpretable automaton. It uses the trajectories from automata learning to efficiently train a simple feed-forward neural network by solving well-posed, single-step IVPs. During inference, the implementation detects the events and solves the IVP piecewise. Preliminary empirical results show significant improvements in training time and computational resource requirements while retaining all advantages of a piecewise solution.

## 1 Introduction

Ordinary Differential Equations (ODEs) are widely used to model continuous systems with discrete events. Examples include collisions in physical systems, fast latent processes in biochemical processes (Fröhlich et al., 2016), and discrete jumps or instantaneous dynamic switch in control theory (Ackerson & Fu, 1970). In this context events are time instances at which the system's dynamics are not differentiable. By specifying an event, we can augment the numerical integration process to detect events and terminate before discontinuities occur; e.g., (Hairer et al., 1993; Shampine & Thompson, 2000; Chen et al., 2021). Solving ODEs with discontinuities results in a piecewise solution that is continuous at the event points (Hairer et al., 1993; Ruohonen, 1994).

For piecewise ODEs, we (i) solve the event detection problem (EDP) to detect discontinuities and (ii) learn what happens at the discontinuities. Chen et al. (2018) introduced Neural ODEs (NODEs), where a Neural Network (NN) replaces the ODE. Later, they introduced Neural Event ODEs to learn piecewise ODEs by iteratively solving the EDP using predefined event and state update functions (Chen et al., 2021). We enhance Neural Event ODEs with (i) automata learning to infer both event and state update functions from system's whose ODE is unknown to the learner (subsequently called black-box ODEs), and (ii) a more efficient training process. Both enhancements ensure well-posed initial value problems (IVPs) per piece.

ODEs with discontinuities are a subclass of Hybrid Automata (HA) that model systems exhibiting both continuous and discrete behaviors (Henzinger, 1996; Poli et al., 2021). In HA, the continuous dynamics are defined by ODEs, and the discrete dynamics and events are defined by jump transitions between them (Henzinger, 1996; Shi & Morris, 2021). Bloem et al. (2020) showed that if the history of computations is not conflicting, the L* algorithm converges to an automaton. Otherwise, we can

split the L\*'s observation table and try to learn a set of automata. In the case of arithmetic black boxes implementing ODEs, the conflicting history of computation arises if the black box is not a function or does not satisfy the realizability conditions described in (Bloem et al., 2020). Following this principle, our first contribution is an automata learning approach that infers the jumps and event specifications from black-box piecewise ODEs.

The proposed automata learning approach yields a high-coverage dataset that captures all events and their effects on the system. Aichernig et al. (2019) showed that such datasets are well-suited to effectively learning surrogate models of hybrid systems. Our second contribution builds on our first contribution and the work of Legaard et al. (2023), who proposed slicing the ground truth trajectory into single-step pairs when training NODEs to ensure it is approximating the next state from a valid previous one. Since the learned event specification slices the trajectory around events, we can extend Legaard et al.'s approach to piecewise ODEs by removing training pairs with a discontinuity between them from the training data. By training the Neural Event ODEs on state pairs in parallel, we eliminate the need for iterating over continuous pieces and repetitive EDP solving. We train the NN once and solve the EDP while learning the state update functions. One could use our first contribution (i) to pre-slice ODEs to handle events with NODEs (Chen et al., 2018) and LatentODEs (Rubanova et al., 2019), or (ii) to validate Neural Event ODEs and LatSegODEs (Shi & Morris, 2021). However, this work mainly focuses on the explainability and applicability of our first contribution to Neural Event ODEs, and we leave its applications, extensions, and numerical comparisons to other methods for future work.

Suppose an ODE $\dot{y} = f(t, y(t), \phi)$, where $y(t)$ is a continuous-time state, $\dot{y} = dy/dt$, $f$ determines how the state changes over time, and $\phi$ is a set of parameters to $f$. We denote by $y(t; t_0, y_0)$, a solution at time t with $y(t_0) = y_0$. Supposing an event at time $t_e$ causes a discontinuity, then:

$$y(t; t_0, y_0) = y_0 + \int_{t_0}^{t_e} f(t, y(t; t_0, y_0), \phi)\, dt + \int_{t_e^+}^{t} f(t, y(t; t_e, y_e), \phi)\, dt, \qquad (1)$$

where $t_e^+$ is infinitesimally greater than time $t_e$ and $y_e = y(t_e; t_0, y_0)$; for more details see (Hairer et al., 1993; Shampine & Thompson, 2000). Meanwhile, the EDP is to find $t_e$ for an event function $g(t, y(t), \psi)$ constrained to be zero at the event point and non-zero elsewhere. For $k$ event functions $g_j(t, y(t), \psi_j)$, an event occurs if the predicate below is satisfied for $j < k$ and $t_e^j$ in the interval $\mathbb{T}$:

$$\forall y(t; t_0, y_0)\, \exists j, t_e^j \,.\, g_j(t_e^j, y(t; t_0, y_0), \psi_j) = 0. \qquad (2)$$

The solution to the EDP is the set $\{t_e^j\}$. In well-posed and decidable IVPs, the EDP is solved with the IVP using event detection algorithms during numerical integration to locate events and terminate if need be (Hairer et al., 1993; Ruohonen, 1994; Shampine & Thompson, 2000; Chen et al., 2021).

The user specifies the events and how they affect the system. The event and state update functions stem from these specifications. We can rewrite both functions as a logical formula over the state and its derivatives. However, writing such functions requires understanding the system's behavior and its events, which is challenging when the system's behavior is unknown (i.e., black box) or complex.

Finally, we chose the bouncing ball as our running example throughout the paper because it encompasses all crucial characteristics of a piecewise dynamical system, including the Zeno behavior, where a system experiences infinite jumps within a finite time interval. This selection emphasizes our method's proficiency in demystifying sophisticated systems into interpretable models.

**Bouncing Ball 1.** We can model a bouncing ball using an ODE of its height $h(t)$ and velocity $v(t)$:

$$f(t, \langle h(t), v(t) \rangle, \{g, e\}) := \begin{cases} \dot{h}(t) = v(t) \\ \dot{v}(t) = -g & \text{if } h(t) > 0 \\ \acute{v}(t) = -ev(t) & \text{if } h(t) = 0 \end{cases} \qquad (3)$$

where $g$ is the gravitational acceleration, $e$ is the elastic coefficient of restitution, and $\acute{v}(t) = v(t^+)$. This piecewise ODE is discontinuous at each bounce when $h(t) = 0$. We can specify this event and its impact on the ball's state by $h(t) = 0 \land v(t) < 0 \rightarrow \acute{v}(t) > 0$.

## 2 LEARNING EVENT SPECIFICATIONS USING AUTOMATA LEARNING

This section presents a method to learn event specifications from a black-box ODE. We use automata learning to infer an abstract model of the ODE and learn the event specifications from it.

Let $\Sigma$ and $\Gamma$ be two disjoint alphabets. A word $\upsilon$ over $\Sigma$ is a string of symbols from $\Sigma$. A word $\omega$ over $\Sigma \times \Gamma$ is a bi-word. A language over $\Sigma \times \Gamma$ is a bi-language. Given $\upsilon = \sigma_1 \sigma_2 \ldots \sigma_n$ in $\Sigma^*$ and $\mu = \gamma_1 \gamma_2 \ldots \gamma_n$ in $\Gamma^*$, we define $\upsilon \oplus \mu = \langle \sigma_1, \gamma_1 \rangle \langle \sigma_2, \gamma_2 \rangle \ldots \langle \sigma_n, \gamma_n \rangle$. We denote the projection of $\omega$ onto $\Sigma$ by $\Pi_\Sigma(\omega) \in \Sigma^*$. Thus, $\Pi_\Sigma(L) = \{\Pi_\Sigma(\omega) \mid \omega \in L\}$ for a bi-language $L$. $L$ is $\Sigma$-exhaustive if $\Pi_\Sigma(L) = \Sigma^*$. $L$ is $\Sigma$-prefix-closed if $\Pi_\Sigma(L)$ is prefix-closed.

**Definition 1** (Mealy Machine). This is a tuple $\mathcal{M} = \langle \Sigma, \Gamma, Q, q_0, \delta, \lambda \rangle$, where $\Sigma$ and $\Gamma$ are finite alphabets, $Q$ is a finite state set, $q_0 \in Q$ is an initial state, $\delta : Q \times \Sigma \rightarrow Q$ and $\lambda : Q \times \Sigma \rightarrow \Gamma$ are the transition and output functions. We extend $\delta$ and $\lambda$ to words as $\delta^* : Q \times \Sigma^* \rightarrow Q$ and $\lambda^* : Q \times \Sigma^* \rightarrow \Gamma^*$ in the standard way. We define $L(\mathcal{M}) = \{\upsilon \oplus \mu \mid \upsilon \in \Sigma^*, \mu = \lambda^*(q_0, \upsilon)\}$.

A discriminating set $\mathbf{E} \subseteq \Sigma^*$ distinguishes the states of an automaton by observing the automaton's output on these words. The Nerode congruency defines state set $Q$ as the set of all states distinguished by $\Sigma^*$, and is the basis of $L^*$, an active automata learning algorithm (Angluin, 1987). Shahbaz & Groz (2009) extended $L^*$ to Mealy machines by learning $\mathbf{E}$ such that $Q$ is the set of all states distinguished by $\mathbf{E}$, given the output function $\lambda$ of $\mathcal{M}$, based on the following lemma.

**Lemma 1** (Shahbaz & Groz, 2009). *Given $\mathcal{M} = \langle \Sigma, \Gamma, Q, q_0, \delta, \lambda \rangle$, and two states $q_1, q_2 \in Q$, we have that $q_1 = q_2$ iff $\lambda^*(q_1, \upsilon) = \lambda^*(q_2, \upsilon)$ for $\upsilon \in \Sigma^*$.*

Given an $\mathcal{M}$, a corresponding $\mathbf{E} \in \Sigma^*$, and two states $q_1, q_2 \in Q$, we say that $\mathbf{E}$ distinguishes $q_1$ and $q_2$ if $\exists \upsilon \in \mathbf{E} : \lambda^*(q_1, \upsilon) \neq \lambda^*(q_2, \upsilon)$. For more details on $L^*$, see (Vaandrager, 2017; Fisman, 2018).

The Nerode congruency relation for bi-languages is defined as follows.

**Definition 2** (Bloem et al., 2020). Given a $\Sigma$-exhaustive bi-language $L$, the relation $\upsilon_1 \sim_L \upsilon_2$ for $\upsilon_1, \upsilon_2 \in \Sigma^*$ is defined by:

$$(\upsilon_1 \sim_L \upsilon_2) \coloneqq (\upsilon_1 \oplus \mu_1) \cdot \omega \in L \text{ iff } (\upsilon_2 \oplus \mu_2) \cdot \omega \in L \text{ for all } \mu_1, \mu_2 \in \Pi_\Gamma(L), \omega \in (\Sigma \times \Gamma)^*$$

$L^*$ and its extensions only terminate if the target language is over finite alphabets with finitely many congruencies. Abstract automata learning extends $L^*$ to learn an abstract model of target languages over large or infinite alphabets, bounding the state-space of learned automata when dealing with infinitely many congruencies; e.g., see (Aarts et al., 2012; Howar et al., 2011). It also learns a correct transition function from finitely many congruencies over infinite alphabets; e.g., see (Mens & Maler, 2015; Maler & Mens, 2017; Drews & D'Antoni, 2017; Moerman et al., 2017). $L^*$ has been extended to black-boxes with timed behaviors; e.g., Mealy machines with timers (Vaandrager et al., 2023), and timed automata (Tang et al., 2022; An et al., 2020). Our work differs as we abstract a function of continuous time by projecting it onto a dense time domain. Abstraction handles large or infinite alphabets obtained from the function's domain and co-domain. We apply the abstraction layer to the extension of $L^*$ for Mealy machines (Shahbaz & Groz, 2009). This allows us to infer an abstract model of the function and learn its event specifications.

## 2.1 Learning Event Specifications

By quantizing continuous time to dense time, we define an input exhaustive and prefix-closed bi-language corresponding to a function of time that we can learn by automata learning.

### 2.1.1 A Dense Model for Functions of Continuous Time

In a continuous time interval $\mathbb{T} = [t_0, t_k]$, a *time point* is a real number $t \in \mathbb{T}$, and a *time sequence* is a sequence of time points $w_t = t_0 t_1 \ldots t_n$ where $t_i < t_{i+1}$. The absolute difference between any two $t_n, t_m \in \mathbb{T}$, denoted by $\tau = |t_m - t_n|$, is referred to as a timestep. Given a set of timesteps $\mathbf{T} = \{\tau_1, \tau_2, \ldots, \tau_n\}$ where each $\tau_i \in \mathbb{R}_+$ and $t_0 + \tau_i \leq t_k$, we define an ordered-set $\mathbb{T}_{\mathbf{T}}$ as all time points in $\mathbb{T}$ using timesteps from the set $\mathbf{T}$, as follows:

$$(\mathbb{T}_{\mathbf{T}}, <) = \bigcup \left( \{(t_0 + n\tau) \quad \text{for} \quad 0 \leq n \leq \lfloor (t_k - t_0) \div \tau \rfloor\}, < \right) \quad \text{for} \quad \tau \in \mathbf{T} .$$

We denote $\mathbf{T} \cup \{0\}$ as $\mathbf{T}$. We define $\mathbf{T}^*$ with respect to $\mathbb{T}$ as follows:

$$\mathbf{T}_{\mathbb{T}}^* = \left\{ \tau_1 \tau_2 \ldots \tau_n \mid t_0 + \sum\nolimits_{i=1}^{n} \tau_i \leq t_k \quad \text{for} \quad n \geq 0 \right\} .$$

Given $w_\tau = \tau_1 \tau_2 \ldots \tau_n$ we use $t_0 + w_\tau$ to denote $w_t = t_0 t_1 \ldots t_n$ such that $t_{i>0} = t_{i-1} + \tau_i$. For $w_\tau = \tau_1 \tau_2 \ldots \tau_n$ and $w_\tau' = \tau_1' \tau_2' \ldots \tau_m'$, we define $w_\tau < w_\tau' \iff (\sum_{i=1}^{n} \tau_i) < (\sum_{i=1}^{m} \tau_i')$. Finally, we define $\mathbb{T}_{\mathbf{T}}^*$ as the set of all $w_t$ that can be generated from $\mathbf{T}_{\mathbb{T}}^*$ as follows:

$$(\mathbb{T}_{\mathbf{T}}^*, <) = \{t_0 + w_\tau \mid w_\tau \in (\mathbf{T}_{\mathbb{T}}^*, <)\} .$$

Given $f : \mathbb{T} \to \mathbb{R}$, we define $f_{\mathbf{T}} : \mathbb{T}_{\mathbf{T}} \to \mathbb{R}$ and generalize it to $f_{\mathbf{T}}^* : \mathbb{T}_{\mathbf{T}}^* \to \mathbb{R}^*$ as follows:
$$f^*(\epsilon) = \epsilon \quad \text{and} \quad f^*(\mathrm{t}_0 \mathrm{t}_1 \ldots \mathrm{t}_n) = f(\mathrm{t}_0) \cdot f^*(\mathrm{t}_1 \ldots \mathrm{t}_n) \,.$$
Accordingly, we define $L(f_{\mathbf{T}})$ as a set of timed words over $\mathbb{T}_{\mathbf{T}} \times \mathbb{R}$ as follows:
$$L(f_{\mathbf{T}}) = \{\mathrm{t}_0 \mathrm{t}_1 \ldots \mathrm{t}_n \oplus f^*(\mathrm{t}_0 \mathrm{t}_1 \ldots \mathrm{t}_n) \mid \mathrm{t}_0 \mathrm{t}_1 \ldots \mathrm{t}_n \in \mathbb{T}_{\mathbf{T}}^*\} \,.$$
Since $L(f_{\mathbf{T}})$ is exhaustive and prefix-closed, we model $L(f_{\mathbf{T}})$ using $\mathcal{M}_f = \langle \mathbf{T}, \mathbb{R}, \mathbb{T}_{\mathbf{T}}, \mathrm{t}_0, \delta, \lambda \rangle$ with $\delta(\mathrm{t}, \tau) = \mathrm{t} + \tau$ and $\lambda(\mathrm{t}, \tau) = f(\mathrm{t} + \tau)$. Finally, if $\mathbb{T}_{\mathbf{T}}$ is finite, then $\mathcal{M}_f$ is a finite Mealy machine that partitions $L(f_{\mathbf{T}})$ into finitely many congruencies. Depending on the quantization granularity, the above construction can result in a huge state space for $\mathcal{M}_f$, making it infeasible to learn.

### 2.1.2 ABSTRACTING FUNCTIONS OF DENSE TIME

By using predicates to partition the alphabets of $\mathcal{M}_f$ into finite subsets, we reduce $\mathcal{M}_f$'s state space considerably. We define an abstraction layer for $L(f)$, enabling efficient automata learning.

A predicate $\vartheta$ over $\mathrm{rng}(f)$ is a function $\vartheta_f : \mathrm{rng}(f) \to \mathbb{X}$, where $\mathbb{X}$ is a discrete finite domain. We denote the domain of all predicates over $\mathrm{rng}(f)$ using $\Theta_f$. We use $\vartheta$ as shorthand for $\vartheta_f$ when $f$ is clear from the context. Given a predicate $\vartheta$, a sequence $r_0 r_1 \ldots r_n$ over $\mathrm{rng}(f)$, we define $\vartheta^*$ by:
$$\vartheta^*(\epsilon) = \epsilon \quad \text{and} \quad \vartheta^*(r_0 r_1 \ldots r_n) = \vartheta(r_0) \cdot \vartheta^*(r_1 \ldots r_n) \,.$$
Similarly, given $f : \mathbb{T} \to \mathbb{R}$, we generalize $\vartheta_f^*$ over a sequence of time points $\mathrm{t}_0 \mathrm{t}_1 \ldots \mathrm{t}_n$ as follows:
$$\vartheta_f^*(\epsilon) = \epsilon \quad \text{and} \quad \vartheta_f^*(\mathrm{t}_0 \mathrm{t}_1 \ldots \mathrm{t}_n) = \vartheta(f(\mathrm{t}_0)) \cdot \vartheta_f^*(\mathrm{t}_1 \ldots \mathrm{t}_n) \,.$$
We define a predicate change detector $\varrho : \Theta \times \mathbb{T}_{\mathbf{T}}^* \to \mathbb{T}$ as follows:
$$\varrho(\vartheta_f, \mathrm{t}_0 \mathrm{t}_1 \ldots \mathrm{t}_n) = \begin{cases} \mathrm{t}_i & \text{if } \exists i : \vartheta_f(\mathrm{t}_i) \neq \vartheta_f(\mathrm{t}_{i-1}) \text{ and } \forall j < i : \vartheta_f(\mathrm{t}_j) = \vartheta_f(\mathrm{t}_{j-1}) \\ \mathrm{t}_0 & \text{otherwise} \end{cases}$$
That is, given $w_{\mathrm{t}} = \mathrm{t}_0 \mathrm{t}_1 \ldots \mathrm{t}_n \in \mathbb{T}_{\mathbf{T}}^*$ and $\vartheta_f^*(w_{\mathrm{t}}) = \chi_0 \chi_1 \ldots \chi_n$, we define $\varrho(\vartheta_f, w_{\mathrm{t}}) = \mathrm{t}_i$ where $\chi_{i-1} \neq \chi_i$ for the first time along $\vartheta_f^*(w_{\mathrm{t}})$; otherwise, $\varrho(\vartheta_f, w_{\mathrm{t}}) = \mathrm{t}_0$. We extend $\varrho$ to a set of predicates $\Theta$ such that if any of the predicates in $\Theta$ changes, then $\varrho(\Theta_f, w_{\mathrm{t}}) = \varrho(\vartheta_f, w_{\mathrm{t}})$ as follows:
$$\varrho(\Theta_f, \mathrm{t}_0 \mathrm{t}_1 \ldots \mathrm{t}_n) = \begin{cases} \varrho(\vartheta_f, \mathrm{t}_0 \mathrm{t}_1 \ldots \mathrm{t}_n) & \text{if } \exists \vartheta \in \Theta : \varrho(\vartheta_f, \mathrm{t}_0 \mathrm{t}_1 \ldots \mathrm{t}_n) = \mathrm{t}_i \\ & \quad \text{and } \forall \mathrm{t}_j < \mathrm{t}_i, \forall \vartheta' \in \Theta : \varrho(\vartheta'_f, \mathrm{t}_0 \mathrm{t}_1 \ldots \mathrm{t}_j) = \mathrm{t}_0 \\ \mathrm{t}_0 & \text{otherwise} \end{cases}$$

**Definition 3** (Abstract Model of a Function of Continuous Time). Given $f : \mathbb{T} \to \mathbb{R}$, a corresponding $\mathcal{M}_f = \langle \mathbf{T}, \mathbb{R}, \mathbb{T}_{\mathbf{T}}, \mathrm{t}_0, \delta, \lambda \rangle$ and two sets of input and output predicates over $\mathrm{rng}(f)$, denoted by $\Theta$ and $\Xi$, we define an abstract model $\mathcal{A}_f = \langle \Theta, \Xi, \mathbb{T}_{\mathbf{T}}, \mathrm{t}_0, \Delta, \Lambda \rangle$ such that $\Delta : \mathbb{T}_{\mathbf{T}} \times \Theta \to \mathbb{T}_{\mathbf{T}}$ and $\Lambda : \mathbb{T}_{\mathbf{T}} \times \Theta \to \mathbb{X}^n$ with $n = \max(1, |\Xi|)$ are defined as follows:
$$\Delta(\mathrm{t}, \vartheta) = \begin{cases} \varrho(\Xi_\lambda \cup \{\vartheta_\lambda\}, \mathrm{t} + w_\tau) & \text{if } \exists w_\tau \in \mathbf{T}_{\mathbb{T}}^* : \varrho(\Theta_\lambda \cup \Xi_\lambda, \mathrm{t} + w_\tau) = \delta^*(\mathrm{t}, w_\tau) \text{ and} \\ & \quad \forall w'_\tau \in \mathbf{T}_{\mathbb{T}}^* : w'_\tau < w_\tau \implies \varrho(\Theta_\lambda \cup \Xi_\lambda, \mathrm{t} + w'_\tau) = \mathrm{t} \\ \min(\mathrm{t} + \max(\mathbf{T}_{\mathbb{T}}^*), \mathrm{t}_k) & \text{otherwise} \end{cases}$$
$$\Lambda(\mathrm{t}, \vartheta) = \begin{cases} \{\xi \mid \forall \xi \in \Xi : \xi(\lambda \circ \Delta(\mathrm{t}, \vartheta))\} & \text{if } \Xi \neq \emptyset \\ \vartheta(\lambda \circ \Delta(\mathrm{t}, \vartheta)) & \text{otherwise} \end{cases}$$
Similarly, we extend $\Delta$ and $\Lambda$ to words over $\Theta$ and $\Xi$ in the standard way, denoted by $\Delta^*$ and $\Lambda^*$.

Notably, $\mathcal{A}_f$ halts on events in $\Xi$ while pursuing to detect events in $\Theta$. That is, once learned, detected events in $\Xi$ determine the transitions of $\mathcal{A}_f$. Finally, $\mathcal{A}_f$ has a considerably smaller state space.

**Bouncing Ball 2.** We implemented the learner using sympy (Meurer et al., 2017) and aalpy (Muskardin et al., 2022). Given $g = 9.81$, $e = 0.8$, $h_0 = 10$, and $v_0 = 0$, the learner infers the ball's $\mathcal{A}_f$ over $\mathbb{T} = [0, 13)$ with a finite set $\mathbf{T}$. Initially, $\Theta = \{v < 0, v = 0, v > 0\}$ and $\Xi = \emptyset$. A correct automaton is learned at $\mathrm{t} = 2.56$ (refer to Fig. 1a and Fig. 1b). The learner marks detected events along the $h(\mathrm{t})$ and $v(\mathrm{t})$ trajectories, effectively slicing them. To improve interpretability, we redefine $\Theta = \{\mathrm{sign}(v)\}$ and $\Xi = \{h < 0, h = 0, h > 0, v < 0, v = 0, v > 0\}$. The learning time remains at $\mathrm{t} = 2.56$. The learner yields Fig. 1c, revealing different stages along the ball's trajectory: free-fall transition $q_0 \to q_1$, collision $q_1 \to q_2$, rebound $q_2 \to q_3$, rising $q_3 \to q_4$, and the peak height $q_4 \to q_0$. However, it does not capture the ball's resting state where both $h$ and $v$ are zero.

**Zeno behavior.** To capture the ball's resting state, the learner learns the Zeno behavior of its ODE, performing a conformance test until $\mathrm{t} = 12.60$. Figure 1d depicts the automaton discarding the input

labels for better readability. $L^\star$ unravels the previous $\mathcal{A}_f$ nine times to state $q_{45}$, where it detects a free-fall, and collision $q_{45} \rightarrow q_{46} \rightarrow q_{47}$. The ball's height nears zero at t = 11.44 and remains until it rests. The velocity changes until the final free-fall $q_{74} \rightarrow q_{75}$, where the ball rests.

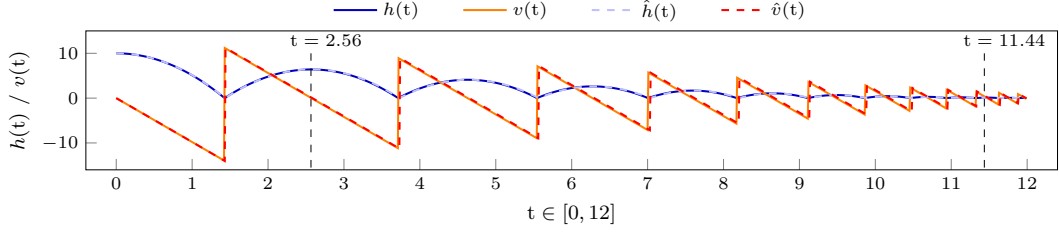

(a) The simulated ground truth vs. predicted dynamics of a bouncing ball until its resting state with a twin y-axis

(b) $\mathcal{A}_f$ w/o output predicates

(c) $\mathcal{A}_f$ with output predicates

(d) $\mathcal{A}_f$ capturing ball's resting state (Zeno behavior)

Figure 1: Learned abstract models of the bouncing ball ODE

### 2.1.3  EVENT EXTRACTION AND SPECIFICATION LEARNING

Although $\mathcal{A}_f$ captures the dynamics governing $f$, it is not easy to interpret. To address this issue, we introduce a systematic approach that learns an explainable structure $\mathcal{S}_f$ resembling $f$'s HA. Initially, we extract significant events from $\mathcal{A}_f$, such as abrupt changes in $f$'s trajectory or its rate of change. These events are then used to learn an abstract event model $\mathcal{E}_f$ that slices $f$ into pieces. Finally, the revealed pieces are merged into similar locations, constructing $\mathcal{S}_f$.

**Function Events.** A function event is characterized by a sign change along two consecutive transitions of a loop-free path in $\mathcal{A}_f$. That is, if $y(q_i)$ and $y(q_{i+1})$ satisfy the same predicates and $\dot{y}(q_i)$ and $\dot{y}(q_{i+1})$ satisfy predicates suggesting an abrupt sign change, then an event occurs at $q_i$, which we can describe by predicates in $\{\vartheta^i\} \cup \Xi^i$ where $\Delta(q_i, \vartheta^i) = q_{i+1}$ and $\Lambda(q_i, \vartheta^i) = \Xi^i$.

**Gradient Events.** An acceleration or deceleration in $y$'s rate of changes can be detected by inspecting three consecutive transitions of a loop-free path in $\mathcal{A}_f$. A gradient event is a pairwise sign change between $\dot{y}(q_{i-1})$, $\dot{y}(q_i)$, and $\dot{y}(q_{i+1})$. For systems demonstrating periodic or oscillatory events, we require that $y(q_{i-1})$ and $y(q_{i+1})$ satisfy the same predicates. Predicates along three transitions that are satisfied by the values of $y(q_i)$ and $\dot{y}(q_i)$ specify the gradient event.

**Abstract Event Models.** An abstract event model $\mathcal{E}_f$ is a Mealy machine whose inputs are event predicates. While learning $\mathcal{E}_f$, the $\mathcal{M}_f$ is unraveled to a state where either an event occurs or some output predicates are satisfied. Event transitions slice $f$ into pieces, and output transitions capture the dynamics of each piece. Finally, we use a look-ahead mechanism, that extends the abstraction layer without affecting the $L^\star$ algorithm (Vaandrager et al., 2023), to determine the state updates of event transitions (i.e., the ODE's behavior after an event).

**Event Specifications.** To merge $f$'s pieces revealed by $\mathcal{E}_f$ into $\mathcal{S}_f$ locations and turn $\mathcal{E}_f$'s output transitions into $\mathcal{S}_f$'s location invariants, we enumerate paths to each event transition from the initial state of $\mathcal{E}_f$ and define a corresponding location in $\mathcal{S}_f$ for each path. The output predicates along each path define location invariants in $\mathcal{S}_f$. Jump conditions are defined using immediate event predicates reached by the path leading to a destination location in $\mathcal{S}_f$, which is recursively constructed by considering the $\mathcal{E}_f$'s destination as a new initial state. The process repeats until all paths are exhausted.

**Bouncing Ball 3.** From the abstract model of the bouncing ball in Figs. 1b and 1c, we extracted:

1. $h = 0$ merges the impact and rebound stages $q_1 \to q_2 \to q_3$, specifying the bounce event.
2. $h > 0 \wedge v = 0$ specifies the peak height at $q_4 \to q_0$, between the rising and free fall stages.

Learning with $\Theta = \{h = 0\}$ yields $\mathcal{E}_f$ in Fig. 2a, where $q_0 \to q_1$ signifies the bounce event with a state update $\acute{v} > 0$, originating from a look-ahead mapper. The transition $q_1 \to q_0$ encapsulates both the rising and free fall stages. Despite the input $h = 0$ for $q_1 \to q_0$, the output is $h > 0 \wedge v > 0$, as $\Delta$ employs $\Xi$ to determine the subsequent state. When an input predicate $\vartheta$ triggers the mapper, the next state is determined by observations satisfying $\Xi \cup \{\vartheta\}$ while seeking a solution to $\vartheta$. Consequently, the ball exits $q_0$ with negative velocity but enters with positive velocity, indicating a missing peak height event, which is captured by the gradient event $h > 0 \wedge v = 0 \to \acute{v} < 0$. Redefining $\Theta = \{h = 0 \wedge v < 0, h > 0 \wedge v = 0\}$ yields $\mathcal{E}_f$ shown in Fig. 2b, whose transitions correspond to the free fall stage $q_0 \to q_1$, bounce event $q_1 \to q_2$, rising stage $q_2 \to q_3$, and peak height event $q_3 \to q_0$.

From the abstract event models of the bouncing ball in Figs. 2a and 2b, we construct the event specifications in Figs. 2d and 2e. Each location's invariants are defined using blue self-loops. For instance, the invariant of $q_0 + q_1$ in Fig. 2e is $h > 0 \wedge v < 0$, the output of $q_0 \to q_1$ in Fig. 2b. Red self-loops are added to ensure input completeness and to represent invalid behavior.

**Zeno behavior.** To model the ball's resting state, we execute the learner using function events (excluding gradient events) and request a conformance test until $\text{t} = 12.60$. The $\mathcal{E}_f$ depicted in Fig. 2c captures the ball's resting state in $q_{19}$ through the self-loop $h = 0 \wedge v = 0$. In this $\mathcal{E}_f$, $q_0 \to q_1$ is initiated by the bounce event $h = 0$, and $q_1 \to q_2$ represents the continuous progression of the ball's height, followed by another bounce event $q_2 \to q_3$. The $\mathcal{S}_f$ depicted in Fig. 2f begins at a location corresponding to $q_0$ with a jump corresponding to the bounce event $q_0 \to q_1$ to a location merging $\{q_1 + q_2\}$. The location $q_0$ in $\mathcal{S}_f$ does not allow for a rising stage, as for the ball to rise, it must bounce first. The $\mathcal{S}_f$ captures ten bounces until the ball rests in $q_{19}$ with the invariant $h = 0 \wedge v = 0$.

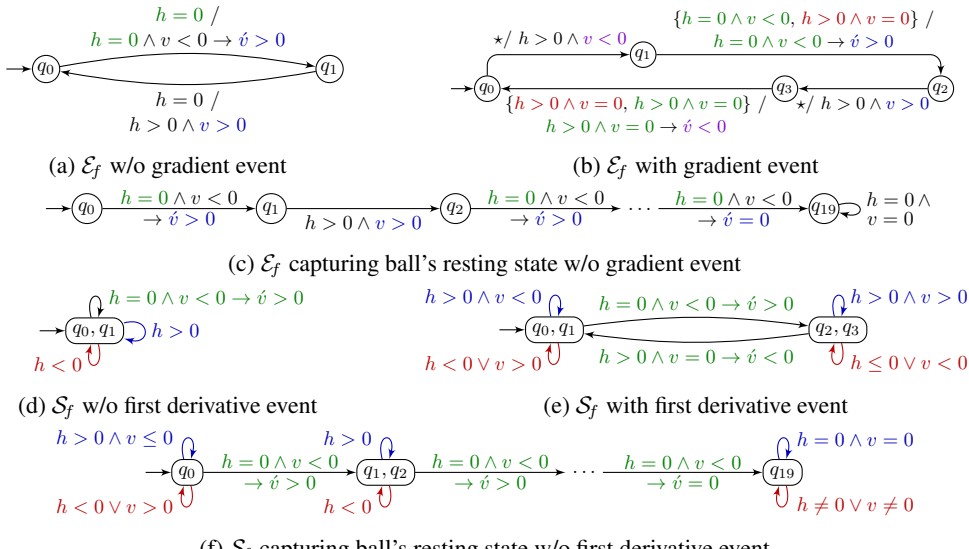

(a) $\mathcal{E}_f$ w/o gradient event

(b) $\mathcal{E}_f$ with gradient event

(c) $\mathcal{E}_f$ capturing ball's resting state w/o gradient event

(d) $\mathcal{S}_f$ w/o first derivative event

(e) $\mathcal{S}_f$ with first derivative event

(f) $\mathcal{S}_f$ capturing ball's resting state w/o first derivative event

Figure 2: Learned event specifications of the bouncing ball ODE using learned abstract models

Appendix A.1 demonstrates how specification learning generalizes to multi-variable systems such as Switching Linear Dynamical Systems (SLDSs).

## 2.2 WELL-POSEDNESS OF THE EVENT SPECIFICATION LEARNING

Given a piecewise ODE $f(\text{t}, y(\text{t}), \phi)$ and an event function $g(\text{t}, y(\text{t}), \psi)$, the isolate occurrence of the event at $\text{t}_e$ slices $f$ into two segments $f_L, f_R$. Shampine & Thompson (2000) showed that, if $y_0$ is well-posed when solving $f$ on a time interval $\mathbb{T} = [\text{t}_0, \text{t}_k]$, then small changes in $y_0$ result in small changes in $y(\text{t} \leq \text{t}_e; \text{t}_0, y_0)$. That is, the solution to $f_L$ varies slightly w.r.t. variations in $\text{t}_0$ and $y_0$ (Coddington & Levinson, 1984). Solving $f_R$ on the remaining interval $(\text{t}_e, \text{t}_k]$ with a perturbed

initial value $y(\mathrm{t} > \mathrm{t}_e; \mathrm{t}_e, y_e)$ is a well-posed problem only if $\mathrm{t}_e$ is a simple root. By repeating the argument, the solution to $f_R$ is also well-posed w.r.t. variations in $\mathrm{t}_e$ and $y_e$. The same applies to several well-separated events in an interval $\mathbb{T}$. For more details, see (Shampine & Thompson, 2000).

Based on this principle, event specification learning reduces to inferring $\mathcal{A}_f$ for a given $f(\mathrm{t}, y(\mathrm{t}), \phi)$. The abstraction from $\mathcal{M}_f$ to $\mathcal{A}_f$ is based on an event detection algorithm for Runge-Kutta method (Hairer et al., 1993, Algorithm 6.4, p. 195). However, instead of a sign change detector, we use $\Delta$ as defined in Definition 3. Given a well-posed $y_0$, and an input predicate $\vartheta$, the $\Delta$ simulates $\mathcal{M}_f$ through $\mathbb{T}_{\mathbf{T}}$ until a predicate in $\Theta \cup \Xi$ is satisfied, or it reaches $\mathrm{t}_0 + \max(\mathbf{T}_{\mathbb{T}}^*, <)$. If we consider each occurrence of $\vartheta \in \Theta$ along the $f$'s trajectory as an event, then $\Xi$ is not necessary for learning $\mathcal{A}_f$. For simplicity, we first describe the case of $\Xi = \emptyset$: In case of an event, $\mathcal{M}_f$ has reached the smallest $\mathrm{t}_e = \delta^*(\mathrm{t}_0, w_\tau)$ that satisfies $\varrho(\Theta_\lambda, \mathrm{t}_0 + w_\tau) = \mathrm{t}_e$ for some $w_\tau \in \mathbf{T}_{\mathbb{T}}^*$. If $\varrho(\vartheta_\lambda, \mathrm{t}_0 + w_\tau) = \mathrm{t}_e$, then $\mathrm{t}_e$ is an isolated occurrence of $\vartheta$ that slices $f$ into $f_L, f_R$ with $\mathrm{t}_e$ being the start of $f_R$ and $\lambda(\mathrm{t}_e)$ being a well-posed initial value for $f_R$. For the case of $\Xi \neq \emptyset$, if we have that $\varrho(\Xi_\lambda, \mathrm{t}_0 + w_\tau) = \mathrm{t}_e$ for some $w_\tau \in \mathbf{T}_{\mathbb{T}}^*$, then $\Delta$ slices $f$ into $f_L$ and $f_R$ regardless of $\vartheta$ with $\lambda(\mathrm{t}_e)$ being a well-posed initial value for $f_R$ at $\mathrm{t}_e = \Delta(\mathrm{t}_0, \vartheta)$. Conversely, if $\varrho(\Theta_\lambda, \mathrm{t}_0 + w_\tau) = \mathrm{t}_e$, we have the above case. This procedure conforms to $k$ event detection predicate shown in Eq. (2).

Given a query $\upsilon \in \Theta^*$, the $\Delta$ iteratively slices $f$; i.e., $\Delta^*(\mathrm{t}_0, \upsilon)$. That is, for the first step of $\Delta^*(\mathrm{t}_0, \upsilon)$ slices $f$ and thereupon it iteratively slices the last $f_R$ by simulating $\mathcal{M}_f$ through $\mathbb{T}_{\mathbf{T}}$ until an isolated occurrence of $\vartheta_n$ at position $n + 1$ along $\upsilon$ or an isolate occurrence of a $\xi \in \Xi$. Finally, $\mathcal{A}_f$ returns $\Lambda^*(\mathrm{t}_0, \upsilon)$ as the answer to $\upsilon$. If no predicate is satisfied along $f$'s trajectory, then $\mathcal{M}_f$ is simulated through $\mathbb{T}_{\mathbf{T}}$ until $\mathrm{t}_0 + \max(\mathbf{T}_{\mathbb{T}}^*, <)$ and $f$ will not be sliced. This is however not a problem as we require $\Theta$ and $\Xi$ to be an over-approximation of the possible changes in the $f$'s trajectory. On the other hand, over-approximating events will not affect the correctness of ODE's solution, for each slice of $f$ is a well-posed IVP and the solution to $f$ is continuous w.r.t. variations in $\mathrm{t}_0$ and $y_0$; see above discussion. This ensures the abstraction layer treats single-piece ODEs correctly.

# 3 LEARNING PIECEWISE DYNAMICS

In NODEs (Chen et al., 2018), we train a NN denoted by $\mathcal{N}(\mathrm{t}, y(\mathrm{t}), \phi)$ on trajectories from the black box ODE of interest. Dealing with piecewise ODEs, it is beneficial to train $\mathcal{N}$ on continuous trajectories in between discontinuities. Thus, $\mathcal{N}$ avoids learning a solution that fits both the continuous dynamics and the discontinuities. However, this implies that we need to use other mechanisms to (i) solve the EDP and (ii) learn the instantaneous state update functions. In this section, we study how to efficiently train NODEs to learn both mechanisms.

## 3.1 LEARNING CONTINUOUS DYNAMICS

Given an initial $y_0$ and a time step $\tau$, we denote $\mathrm{t}_i = \mathrm{t}_0 + i\tau$ and $y_i = y(\mathrm{t}_i)$. Suppose a NN denoted as $\mathcal{N} : \mathbb{T} \times \mathrm{rng}(y) \times \mathrm{dom}(\phi) \to \mathrm{rng}(\dot{y})$ such that $\dot{y}_i = \mathcal{N}(\mathrm{t}_i, y_{i-1}, \phi)$. Starting with $\hat{y}_0 = y_0$, we can predict the next states through the recursive invocation of $\mathcal{N}$; that is, $\hat{y}_{i>0} = \hat{y}_{i-1} + \int \mathcal{N}(\mathrm{t}_i, \hat{y}_{i-1}, \phi)$. In this setup, except $y_0$, a well-posed initial value while inferring the next state is not guaranteed and the error accumulates over time. $\mathcal{N}$ generally tries to compensate for this error, from a future state $\hat{y}_i$ onwards, with a series of incorrect and error rectifying mappings; see (Legaard et al., 2023). Alternatively, given a ground truth trajectory, Legaard et al. (2023) proposed to train $\mathcal{N}$ on single-step pairs. Given $y_0$ and a time step $\tau$, we have $\hat{y}_{i>0} = y_{i-1} + \int \mathcal{N}(\mathrm{t}_i, y_{i-1}, \phi)$. This effectively makes $\hat{y}_i$ a function of $y_{i-1}$ which is guaranteed to be a well-posed initial value. Moreover, by avoiding recursive invocations, we can train $\mathcal{N}$ in parallel on multiple single-step pairs. Since this approach still invokes an ODE solver on single-step pairs, extending it to variable time steps is straightforward.

## 3.2 LEARNING INSTANTANEOUS STATE UPDATES

However, the above training approach is not directly applicable to piecewise ODEs. Suppose an event occurs at $\mathrm{t}_e$ causing an instantaneous change in the state from $y_e$ to $\acute{y}_e$. Then, $\mathcal{N}$ must learn two different mappings: (i) continuous dynamics $\hat{y}_{e+1} = \acute{y}_e + \int \mathcal{N}(\mathrm{t}_e, \acute{y}_e, \phi)$, and (ii) instantaneous state update function (i.e. $y_e \to \acute{y}_e$). Since $y_e$ and $\acute{y}_e$ are of different dynamics, it is difficult to generalize $\mathcal{N}$'s mapping over both dynamics. Chen et al. (2021) proposed to learn these mappings separately. By eliminating successive pairs crossing a discontinuity, we can apply single-step training. This modified method retains the well-posedness of the automata-generated training data; see Section 2.2. After learning the continuous dynamics and receiving state updates from the event specification, we

must learn an exact mapping for instantaneous state update functions. This is achieved by training a nonlinear transformation $\mathcal{U}(t, y(t), \psi)$ derived from the event specification. Training $\mathcal{U}$ reduces to learning the parameters $\psi$ from trajectories between pairs crossing discontinuities.

**Bouncing Ball 4** (Learning Piecewise Dynamics). $\mathcal{N}(t, y(t), \phi)$ has three fully connected layers of 64 units with GELU activation. For the optimizer, we use $\mathrm{Adam}$ with a learning rate of $0.001$. We derive the instantaneous state update function as velocity's sign inversion from the event specification. To estimate $e$ in Eq. (3), we use a trainable non-linear transformation $\mathcal{U}(t, y(t), \psi)$ with a sigmoid activation. We use $\mathrm{Xavier}$ initializer for $\mathcal{N}$, and pytorch's default initializer for $\mathcal{U}$.

Using the dataset we generate while learning $\mathcal{S}_f$, we train $\mathcal{N}$ on 3169 single-step continuous pairs from the interval $\mathbb{T} = [0, 10.43]$ for 5000 epochs. Then we freeze $\mathcal{N}$ and train $\mathcal{U}$ on data points from eight event locations, i.e. slices that go over the discontinuities, for 2000 epochs. The slices around the event locations contain 3, 4, 5, 6, 8, 11, 15, and 21 data points. For both networks, we use MAPE as the training loss and MSE as the validation loss. We deliberately chose MSE for validation to highlight the outlier predictions, making the validation loss significantly higher than the training loss.

For validation, we only provide the model with $y_0 = 10\,\mathrm{m}$ at $t_0$ and $\mathbf{T} = \{0.01\}$. We predict the ball's trajectory until it rests, i.e., $\mathbb{T} = [0, 12)$. The baseline is a ground truth sampled from the black box ODE with $\mathbf{T} = \{0.01\}$. Running experiments with five random seeds resulted in an average loss of $1.14\,\mathrm{m}^2 \pm 0.05\,\mathrm{m}^2$. Figure 1a already shows a predicted trajectory vs. the ground truth.

**On Importance of Hyperparameters.** The architecture and activation function can significantly impact the performance. Given such a shallow architecture for $\mathcal{N}$, we expect the performance to be sensitive to the number of units in each layer. Reducing layer units to 32 results in an average loss of $1.26\,\mathrm{m}^2 \pm 1.05\,\mathrm{m}^2$, which is expected for such a small network. Reducing the number of hidden layers in $\mathcal{N}$ to two results in an average loss of $1.49\,\mathrm{m}^2 \pm 0.76\,\mathrm{m}^2$. Substituting GELU with ReLU in $\mathcal{N}$ results in an average loss of $1.05\,\mathrm{m}^2 \pm 0.44\,\mathrm{m}^2$ showing no significant difference.

**Example** (Thermostat-Controlled Storage-Tank Water Heater). The thermostat turns the heater on when the water temperature reaches a set point $T_{on} = 45\,°\mathrm{C}$, and turns it off upon reaching $T_{off} = 100\,°\mathrm{C}$. The specification learning process is similar to that of the bouncing ball. Assuming the initial temperature $T_0 = T_{on}$, the $\mathcal{S}_f$ is shown in Fig. 3a. This $\mathcal{S}_f$ comprises a range-based predicate, i.e., $T \in [T_{on}, T_{off}]$, demonstrating that we can specify more complex events, such as threshold crossings, common in control systems. Another application of range-based predicates is to specify the *guard conditions*; e.g., dealing with noisy variables in real-world systems. See Appendix A.2 for the ODE, the NN architectures, the training setup, and hyperparameter tuning.

For validation, we only provide the model with $T_0 = T_{on}^-$ at $t_0$, $\mathbf{T} = \{5\}$, and predict the system's dynamics within $\mathbb{T} = [0, 4500]$. The validation baseline is a trajectory sampled from the black box ODE with $\mathbf{T} = \{5\}$. Running the experiments with five random seeds resulted in an average loss of $0.00027 \pm 6.860 \times 10^{-6}$. Figure 3b shows a predicted trajectory vs. the ground truth.

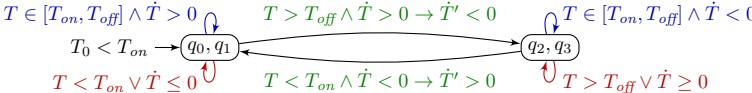

(a) The $\mathcal{S}_f$ with blue self-loops defining location invariants, and red self-loops revealing invalid behaviors.

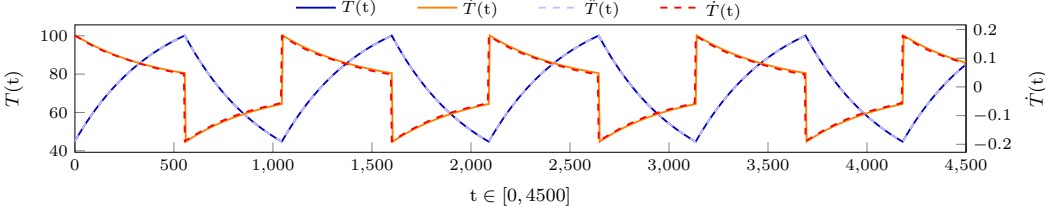

(b) The ground truth vs. the predicted dynamics of the storage-tank water heater with thermostat control

Figure 3: Learning the storage-tank water heater with thermostat control.

## 4    RELATED WORK

Despite accurate results, Neural Event ODEs require extensive prior knowledge about the events and are computationally costly due to repetitive EDP solving on the ODE's complete trajectory. Our enhanced variation does not require prior knowledge about the events and solves the EDP on a subset of the training data only if it is required while learning the parameters of the instantaneous state update function. Not having to solve the EDP and using a single-step training method to learn the continuous dynamics significantly reduces the computational cost of our approach. Finally, we provide an automaton that helps users interpret the system's dynamics.

Simultaneously, Shi & Morris (2021) proposed LatSegODEs, using a changepoint detector (e.g. PELT by Killick et al., 2012), to slice the ODE's trajectory and learn its piecewise dynamics by LatentODEs. LatentODEs (Rubanova et al., 2019), model the latent dynamics of irregularly sampled time series by combining NODEs with Variational Auto Encoders (VAEs) (Kingma & Welling, 2014). Subsequently, LatSegODEs require simple dynamics within each piece for accurate latent space construction. Moreover, constructing a latent space can reduce interpretability due to complex transformations for high-dimensional spaces or the potential loss of essential data in lower dimensions. Our approach differs from LatSegODEs in (i) our approach is more interpretable as it does not construct a latent space and (ii) we do not require a changepoint detector to solve the EDP.

Following the same line of research, Poli et al. (2021) proposed Neural Hybrid Automata (NHA) to model Stochastic Hybrid Systems (SHSs) without prior knowledge of dynamic pieces and events. NHA comprise three modules: (i) a dynamic module, (ii) a discrete latent selector module, and (iii) an event module. The dynamic module is a NODE modeling the continuous dynamics in each SHS mode. The discrete latent selector uses a Normalizing Flow Network (NFN) by Durkan et al. (2019), that given SHS's current mode, identifies a corresponding latent state. Once the latent state is identified, the event module detects an event occurrence and its instantaneous effect on the SHS, updating its mode and the latent state. The precision of the discrete latent selector is enhanced by the accurate modeling of latent states distribution by NFNs, offering better guarantees for transitioning between piecewise dynamics than VAEs-based methods like LatSegODEs. However, due to their complexity, NHA are less interpretable compared to our approach.

## 5    CONCLUSION & FUTURE WORK

This paper presents a hybrid comprehensive approach for inferring an interpretable specification of a system showing piecewise dynamics. We used automata learning to infer an abstract model of a possibly black-box system's behavior and a neural network to learn its continuous dynamics. Automata learning is polynomial in the size of inputs and the number of congruency classes in the target language. Specification learning is of polynomial complexity in the number of input predicates, and congruent events. This is affordable for many real-world systems and allows us to learn an interpretable model of their behavior without prior knowledge.

Next, we demonstrated a more effective training scheme for NNs learning continuous dynamics in the presence of discontinuities that we can extend to other methods such as NODEs and LatentODEs. We should note that we ensured the IVP's well-posedness during automata learning and neural network training, making our approach theoretically sound. Through a step-by-step analysis using the bouncing ball, we demonstrated that our approach can efficiently learn interpretable models of piecewise dynamics with significantly fewer data points and computational resources compared to current state-of-the-art methods. Experimental results on the water heater and the SLDS showed that our approach can learn explainable specifications of complex systems with piecewise dynamics.

For future work, we first aim to incorporate changepoint detection into the automata learning process to augment the predicate change detector. Next, we aim to complement NODEs and LatentODEs with our specification learning approach and perform a thorough numerical comparison with other methods like Neural Event ODEs and LatSegODEs. Applying our approach to LatSegODEs (Shi & Morris, 2021) results in a latent event specification whose conformance check against the event specification of the original system possibly verifies the correctness of the latent space construction. We can also apply our approach to learn interpretable models of physical systems with decomposable dynamics through a compositional approach to automata learning introduced in (Moerman, 2018). This would allow us to apply our approach to large-scale systems with multiple trajectories.

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

# A    LEANING PHYSICAL SYSTEMS

## A.1    A SWITCHING LINEAR DYNAMICAL SYSTEM

Chen et al. (2021) showed how to learn Switching Linear Dynamical Systems (SLDSs) using Neural Event ODEs. SLDSs are of particular interest for extracting interpretable models from physical systems (Ackerson & Fu, 1970; Chang & Athans, 1978), neuroscience (Linderman et al., 2017) or finance (Fox et al., 2008). Although Neural Event ODEs successfully model SLDSs, training them requires the underlying ODE to be known apriori. Using our proposed event specification learning approach, we can learn an explainable model from a black-box SLDS without knowing its ODE. Let us look at the following example. Consider a black-box system that implements the following ODE:

$$f(\text{t}, x(\text{t}), \phi) = \begin{cases} \dot{x} = xA + \begin{bmatrix} 0 & 2 \end{bmatrix} & \text{if } x_1 \geq 2 \\ \dot{x} = \begin{bmatrix} -1 & -1 \end{bmatrix} & \text{if } x_0 \geq 2 \wedge x_1 < 2 \\ \dot{x} = \begin{bmatrix} -1 & +1 \end{bmatrix} & \text{if } x_0 < 2 \wedge x_1 < 2 \end{cases} \quad \text{with} \quad A = \begin{bmatrix} 0 & 1 \\ -1 & 0 \end{bmatrix}, \quad (4)$$

where $A$ is called a rotation matrix, $x \in \mathbb{R}^2$ such that $x = \begin{bmatrix} x_0 & x_1 \end{bmatrix}$ with the initial value $x_0 = \begin{bmatrix} 0 & 0 \end{bmatrix}$. In the following, we show how to learn the event specification of the SLDS and explain its behavior from its black-box implementation without prior knowledge of its ODE.

### A.1.1    EVENT SPECIFICATION

We define $\Sigma = \{\text{sign}(\dot{x})\}$ and $\Gamma = \{x_i \geq 2, x_i < 2, \dot{x}_i < 0, \dot{x}_i = 0, \dot{x}_i > 0\}_{i=0}^1$ where $\text{sign}(\dot{x})$ return a vector of signs of $\dot{x}$. Figure 4a shows the learned abstract function $\mathcal{A}_f$ of the SLDS. Inspecting the $\mathcal{A}_f$, we can extract the following events:

1. $x_0 < 2 \wedge x_1 \geq 2$ with state update $\dot{x}_0 < 0 \rightarrow \dot{x}_0' > 0$
2. $x_0 \geq 2 \wedge x_1 \geq 2$ with state update $\dot{x}_1 > 0 \rightarrow \dot{x}_1' < 0$
3. $x_0 \geq 2 \wedge x_1 < 2$ with state update $\dot{x}_0 > 0 \rightarrow \dot{x}_0' < 0$
4. $x_0 < 2 \wedge x_1 < 2$ with state update $\dot{x}_1 < 0 \rightarrow \dot{x}_1' > 0$

Updating $\Sigma$, learning the abstract event model yields the $\mathcal{E}_f$ shown in Fig. 4b. In the $\mathcal{E}_f$, the transitions $s_0 \rightarrow s_1 \rightarrow s_2$ of the $\mathcal{A}_f$ are combined into $q_0 \rightarrow q_1$ with a dynamic update $\dot{x}_0 < 0 \rightarrow \dot{x}_0' > 0$. Next, $s_2 \rightarrow s_3 \rightarrow s_4$ from the $\mathcal{A}_f$ are merged into $q_1 \rightarrow q_2$ with a dynamic update $\dot{x}_1 > 0 \rightarrow \dot{x}_1' < 0$. Similarly, $s_4 \rightarrow s_5 \rightarrow s_6$ and $s_6 \rightarrow s_7 \rightarrow s_0$ are merged into $q_2 \rightarrow q_3$ and $q_3 \rightarrow q_0$ respectively. The learned $\mathcal{E}_f$ yields the $\mathcal{S}_f$ shown in Fig. 4c that conforms to the definition of SLDS's ODE. Of note, the initial value $x_0 = \begin{bmatrix} 0 & 0 \end{bmatrix}$ determines the initial location $q_0$ of the $\mathcal{S}_f$. For example if $x_0 = \begin{bmatrix} -2 & 2 \end{bmatrix}$, the $\mathcal{S}_f$ would start at $q_1$. Finally, we claim the $\mathcal{S}_f$ provides an intuitive explanation of the SLDS's behavior regardless of whether or not its ODE is known.

### A.1.2    EXPLAINABILITY

Although Eq. (4) has three cases defining the SLDS, our method discovers four locations given the explored trajectory in Fig. 4d. The locations of the learned $\mathcal{S}_f$ correspond to the regions of the SLDS's trajectory in Fig. 4d. The invariant of $q_0$ is $x_0 < 2 \wedge x_1 < 2 \wedge \dot{x}_0 < 0 \wedge \dot{x}_1 > 0$, aligning with the third case of Eq. (4). Initially, the SLDS is in the region corresponding to $q_0$ in Fig. 4d and stays there with $x_0$ linearly decreasing and $x_1$ increasing until $x_1 \geq 2$. Then, according to the first case of Eq. (4), the SLDS starts its rotation and the $\mathcal{S}_f$ jumps to $q_1$ with $\dot{x}_0 < 0 \rightarrow \dot{x}_0' > 0$. The SLDS remains in the $q_1$ region until the progress of $x$ forms an arc reaching $x_0 \geq 2$, causing $\mathcal{S}_f$ to transit to $q_2$, which is also governed by the first case of Eq. (4). In $q_2$ the rotation of $x$ causes a decrease in $x_1$ forming an arc that reaches $x_1 < 2$ in the corresponding region of Fig. 4d, at which point the $\mathcal{S}_f$ jumps to $q_3$ with $\dot{x}_0 > 0 \rightarrow \dot{x}_0' < 0$. Like $q_0$, the values of $x_0$ and $x_1$ progress linearly and proportionally, keeping the $\mathcal{S}_f$ in $q_3$ until $x_0 < 2$ and the $\mathcal{S}_f$ jumps to $q_0$ with $\dot{x}_1 < 0 \rightarrow \dot{x}_1' > 0$.

The $\mathcal{S}_f$'s locations and transitions are learned through a change detector that monitors alterations in the signs of $\dot{x}$ along the given trajectory. This learning process can identify the $x_0$ and $x_1$ values that trigger changes in the SLDS's behavior. For instance, if the SLDS initiates rotation when $x_1 \geq \alpha$, evaluating $x_1$ during the $s_1 \rightarrow s_2$ transition of the $\mathcal{A}_f$ reveals $\alpha = 2$.

### A.1.3    LEARNING PIECEWISE DYNAMICS

For an example of how to use the learned event specification $\mathcal{S}_f$ to model the the continuous dynamics using Neural Event ODEs, we refer the reader to (Chen et al., 2021).

$$x = \begin{bmatrix} 0 & 0 \end{bmatrix} \rightarrow \boxed{s_0}$$
$$\star \, / \, x_0 < 2 \wedge x_1 < 2,$$
$$\dot{x}_0 < 0 \wedge \dot{x}_1 > 0$$

$$\star \, / \, x_0 < 2 \wedge x_1 \geq 2, \quad \boxed{s_1} \quad \star \, / \, x_0 < 2 \wedge x_1 \geq 2, \quad \boxed{s_2} \quad \star \, / \, x_0 \geq 2 \wedge x_1 \geq 2, \quad \boxed{s_3}$$
$$\dot{x}_0 < 0 \wedge \dot{x}_1 > 0 \qquad \dot{x}_0 > 0 \wedge \dot{x}_1 > 0 \qquad \dot{x}_0 > 0 \wedge \dot{x}_1 > 0$$

$$\star \, / \, x_0 \geq 2 \wedge x_1 \geq 2,$$
$$\dot{x}_0 > 0 \wedge \dot{x}_1 < 0$$

$$\boxed{s_7} \quad \star \, / \, x_0 < 2 \wedge x_1 < 2, \quad \boxed{s_6} \quad \star \, / \, x_0 \geq 2 \wedge x_1 < 2, \quad \boxed{s_5} \quad \star \, / \, x_0 \geq 2 \wedge x_1 < 2, \quad \boxed{s_4}$$
$$\dot{x}_0 < 0 \wedge \dot{x}_1 < 0 \qquad \dot{x}_0 < 0 \wedge \dot{x}_1 < 0 \qquad \dot{x}_0 > 0 \wedge \dot{x}_1 < 0$$

(a) $\mathcal{A}_f$ of the SLDS

$$\star \, / \, x_0 < 2 \wedge x_1 \geq 2, \qquad\qquad \star \, / \, x_0 \geq 2 \wedge x_1 \geq 2,$$
$$\dot{x}_0 < 0 \wedge \dot{x}_1 > 0 \rightarrow \dot{x}_0' > 0 \qquad \dot{x}_0 > 0 \wedge \dot{x}_1 > 0 \rightarrow \dot{x}_1' < 0$$

$$x = \begin{bmatrix} 0 & 0 \end{bmatrix} \rightarrow \boxed{q_0} \qquad \boxed{q_1} \qquad \boxed{q_3} \qquad \boxed{q_2}$$

$$\star \, / \, x_0 < 2 \wedge x_1 < 2, \qquad\qquad \star \, / \, x_0 \geq 2 \wedge x_1 < 2,$$
$$\dot{x}_0 < 0 \wedge \dot{x}_1 < 0 \rightarrow \dot{x}_1' > 0 \qquad \dot{x}_0 > 0 \wedge \dot{x}_1 < 0 \rightarrow \dot{x}_0' < 0$$

(b) $\mathcal{E}_f$ of the SLDS

$$x_0 < 2 \wedge x_1 < 2 \wedge \dot{x}_0 < 0 \wedge \dot{x}_1 > 0 \qquad\qquad x_0 < 2 \wedge x_1 \geq 2 \wedge \dot{x}_0 > 0 \wedge \dot{x}_1 > 0$$

$$x_0 \geq 2 \vee \dot{x}_0 \geq 0 \vee \dot{x}_1 \leq 0 \quad \boxed{q_0} \qquad \begin{array}{c} x_0 < 2 \wedge x_1 \geq 2 \wedge \dot{x}_0 < 0 \\ \wedge \dot{x}_1 > 0 \rightarrow \dot{x}_0' > 0 \end{array} \quad \boxed{q_1} \quad x_1 < 2 \vee \dot{x}_0 \leq 0 \vee \dot{x}_1 \leq 0$$

$$x_0 < 2 \wedge x_1 < 2 \wedge \dot{x}_0 < 0 \wedge \dot{x}_1 < 0 \rightarrow \dot{x}_1' > 0 \qquad\qquad\qquad\qquad x_0 \geq 2 \wedge x_1 \geq 2 \wedge \dot{x}_0 > 0 \wedge \dot{x}_1 > 0 \rightarrow \dot{x}_1' < 0$$

$$x_1 \geq 2 \vee \dot{x}_0 \geq 0 \vee \dot{x}_1 \geq 0 \quad \boxed{q_3} \qquad \begin{array}{c} x_0 \geq 2 \wedge x_1 < 2 \wedge \dot{x}_0 > 0 \\ \wedge \dot{x}_1 < 0 \rightarrow \dot{x}_0' < 0 \end{array} \quad \boxed{q_2} \quad x_0 < 2 \vee \dot{x}_0 \leq 0 \vee \dot{x}_1 \geq 0$$

$$x_0 \geq 2 \wedge x_1 < 2 \wedge \dot{x}_0 < 0 \wedge \dot{x}_1 < 0 \qquad\qquad x_0 \geq 2 \wedge x_1 \geq 2 \wedge \dot{x}_0 > 0 \wedge \dot{x}_1 < 0$$

(c) The $\mathcal{S}_f$ with blue self-loops defining location invariants, and red self-loops revealing invalid behaviors.

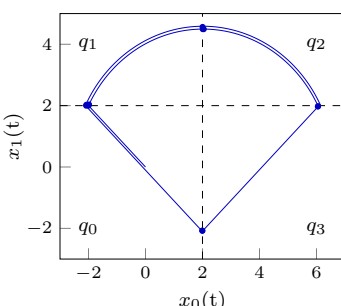

(d) Explored trajectory of the SLDS with dynamic regions labeled to correspond to locations of the $\mathcal{S}_f$.

Figure 4: Learning an event specification of the SLDS.

## A.2 A Thermostat Controlled Storage-Tank Water Heater

The storage tank water heater with an autonomous thermostat forms a piecewise ODE with two events: (i) the heater turns on when the water temperature falls below $T_{on}$, and (ii) it turns off when the temperature rises above $T_{off}$. Given parameters $P$ (heater's power), $U$ (heat transfer coefficient), $m$ (water mass), $C_p$ (water's specific heat capacity), and $T_{amb}$ (ambient temperature), we define $\phi = \{P, U, m, C_p, T_{amb}, T_{on}, T_{off}\}$ and the ODE using $T(t)$ as the water temperature:

$$f(\mathsf{t}, T(\mathsf{t}), \phi) = \begin{cases} \dot{T} = P/mC_p + \ell(\mathbb{T}) & \text{if } T < T_{on} \\ \dot{T} = \ell(T) & \text{if } T > T_{off} \end{cases} \quad \text{with} \quad \ell(T) := -U(T - T_{amb})/mC_p\,,$$

where $\ell(T)$ is the heat loss function. We deliberately chose such a thermostat system as its dynamics are complex enough to challenge ML methods. More specifically, the ODE slice for $T < T_{on}$ resembles the reverse of the slice for $T > T_{off}$. This observation reflects the underlying physical principles governing the heating and cooling processes. However, this physical principle can be unintuitive for NODEs to learn (Ott et al., 2023) since the ODE is not straightforwardly symmetric as it does not satisfy the symmetry condition $f(\mathsf{t}, T(\mathsf{t}), \phi) = f(-\mathsf{t}, T(-\mathsf{t}), \phi)$ for all $\mathsf{t}$. Finally, discussing the Lie group of symmetries of the ODE is beyond the scope of this paper.

### A.2.1 Event Specification

We define $\Sigma = \{\text{sign}(\dot{T})\}$ and $\Gamma = \{T < T_{on}, T > T_{off}, T \in [T_{on}, T_{off}], \dot{T} < 0, \dot{T} > 0\}$ as the input and output alphabets, respectively. Figure 5a shows the learned abstract function $\mathcal{A}_f$ of the thermostat. Inspecting the $\mathcal{A}_f$, we can see a clear separation of the two events; the locations $q_0$ and $q_1$ correspond to the heater being on, and $q_2$ and $q_3$ correspond to the heater being off. Learning with $\Sigma = \{T < T_{on}, T > T_{off}\}$ yields the abstract event model $\mathcal{E}_f$ with an identical structure, as shown in Fig. 5b. Applying the event specification construction on the $\mathcal{E}_f$ yields the $\mathcal{S}_f$ shown in Fig. 3a.

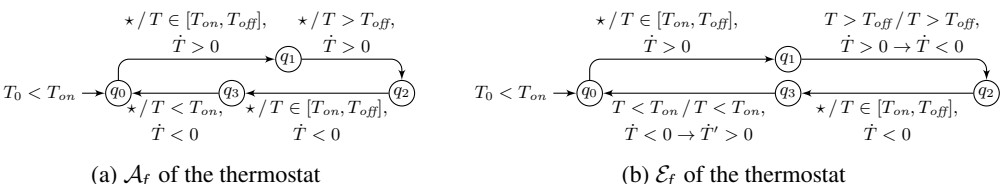

(a) $\mathcal{A}_f$ of the thermostat        (b) $\mathcal{E}_f$ of the thermostat

Figure 5: Learning the event specification of the thermostat.

This example shows we can use range-based predicates, i.e., $T \in [T_{on}, T_{off}]$. Such capability lets us easily specify more complex events, such as threshold crossings, common in control systems. Another application of range-based predicates is to specify the *guard conditions* in case of dealing with noisy variables in real-world systems.

### A.2.2 Learning Piecewise Dynamics

To learn the piecewise dynamics, we convert the learned $\mathcal{S}_f$ to a NHA with two networks $\mathcal{N}_{on}$ and $\mathcal{N}_{off}$ modeling dynamics of the heater when it is on and off, respectively. Both $\mathcal{N}$ are feed-forward NNs with five hidden layers of 64 units and a GELU activation function. Additionally, we use two networks $\mathcal{U}_{on}$ and $\mathcal{U}_{off}$ to update $\dot{T}(\mathsf{t})$ as the $\mathcal{S}_f$ suggests. Both $\mathcal{U}$ are feed-forward NNs with five hidden layers of 512, 128, 64, 32 units, respectively. We learn a task-specific ParametricGELU activation function for $\mathcal{U}$ using the approach in (Basirat & Roth, 2018; 2019). We use pytorch's default initializer, an Adam optimizer with a learning rate of 0.001, and a Huber loss function. We simultanenously train $\mathcal{N}_{on}$ and $\mathcal{N}_{off}$ for 2000 epochs on continuous single-step pairs. Subsequently, we train $\mathcal{U}_{on}$ and $\mathcal{U}_{off}$ for 100 epochs on slices of 121 samples around the events.

**On Importance of Hyperparameters.** Reducing the number of hidden layers in $\mathcal{N}$ to three results in an average loss of $0.060 \pm 0.08$. Reducing the number of units in each layer to 32 results in an average loss of $0.0003 \pm 1.7 \times 10^{-9}$. Substituting the GELU with Mish (Misra, 2020) in $\mathcal{N}$ results in an average loss of $0.0003 \pm 0.0001$ showing no significant difference.

