# OpenReview forum: "Automata Learning for Neural Event ODEs: An Interpretable Model of Piecewise Dynamics"
_ICLR.cc/2024/Conference — Submitted to ICLR 2024_

### Official Review · Reviewer_fYaA · 2023-10-31

**Soundness:** 2 fair
**Presentation:** 3 good
**Contribution:** 2 fair
**Rating:** 3
**Confidence:** 3

**Summary:**

In this paper, authors have presented a hybrid approach for inferring an interpretable specification of a system showing piecewise dynamics. Authors used automata learning to infer an abstract model of a possibly black-box system’s behavior and a neural network to learn its continuous dynamics. Automata learning is polynomial in the size of inputs and the number of congruency classes in the target language. Next, authors demonstrated a more effective training scheme for neural networks learning continuous dynamics in the presence of discontinuities. Through a step-by-step analysis using the bouncing ball, authors demonstrated that the proposed approach can efficiently learn interpretable models of piecewise dynamics with significantly fewer data points and computational resources compared to current state-of-the-art Methods.

**Strengths:**

Compared with previous methods, the authors propose a new method for solving the EDP which does not require prior knowledge about the events and solves the EDP on a subset of training data only if it is required. It also learns an automaton to interpret the system's dynamics. The proposed method is mostly original and potentially is an improvement over previous methods.

The paper is written and organized well. Sufficient previous papers are cited and compared.

**Weaknesses:**

1) This paper is still in its preliminary form. It is shorter, compared with other submissions to a major conference. Authors only demonstrate the evaluations on the dataset of bouncing ball, which is a toy example in neural event ODE.

2) A lot of previous methods are mentioned and cited, but they are not compared numerically compared with the proposed method, such as LatentODE and LatSegODE. Also, ablation studies are missing, and performance comparisons should be conducted for the proposed methods with different choices of important parameters.

3) Authors claim that the proposed method has reduced computational cost. But convincing experimental evidence needs to be shown. Neural event ODE methods always have significant computational cost, including the proposed method, so its comparison with previous methods is needed.

**Questions:**

1) Authors propose to use L* for automaton learning. Why to choose this method? Is there any better choice? The motivation should be presented more.

2) Since the automaton learning algorithms always need a lot of data to cover all the possible prefixes, can authors provide some complexity analysis on the minimum amount of necessary training data?

3) Can the proposed method handle the situation of noisy data? Can the automaton still be learned from noisy data?

4) please provide more experimental evaluation to verify the framework.

5) please include comparison with alternative methods to verify the improvements over state of art.

---

> ### Author Response · Authors · 2023-11-23
>
> Dear Reviewer fYaA,
>
> Thank you for dedicating your time to review our paper. It appears that you have a deep understanding of Neural ODEs (NODEs), but you may not specialize in the field of ODEs, PDEs, and SDEs. This might explain why the comments primarily focus on performance comparisons. We would like to clarify that the main contribution is in the realm of eXplainable AI for ODEs (not SDEs), and additional benefits can be realized through the correct application of the learned automaton. We hope this addresses any potential misunderstandings.
>
> **Regarding weaknesses:**
> 1. Given the page limit, our focus was on the method and discussing the well-posedness of the problem guaranteed by the abstraction layer (Sections 2.1.1 and 2.1.2). We respectfully disagree with the reviewer's perspective. We are confident that our proposed theory is sound, complete, and mature enough to be recognized as a significant contribution.
>
> 2. Indeed, we selected the bouncing ball example due to its intriguing properties, including the Zeno behavior. To our knowledge, no other work captures or describes Zeno behavior. Our examination of the Neural Event ODEs (NEVs) implementation (Chen et al., 2021) confirmed it does not address Zeno. Additionally, we included two examples with real-world implications:\
>     2.1. A thermostat-controlled water heater, showing method's applicability in solving non-trivial Lie symmetric groups, e.g. symmetry of the heating and cooling processes.\
>     2.2. A switching dynamical system, illustrating how to generalize event specification learning to multi-variable black-box systems, and still learn a model that is more explainable compared to their ODEs. Notably, switching dynamical systems are widely used to interpret systems in neuroscience, finance, and physics.
>
> 3. Our primary contribution is the inference of an explainable model for systems implementing piecewise dynamics, without the need for their ODEs (aka. black-box ODEs). This model is immediately interpretable by humans. In contrast, LatentODEs and LatSegODEs use latent space representations that require extensive post-processing to provide some level of insight and interpretability.
> We also assert that our training method for NEVs is more efficient. It's crucial to note that segments of Event ODEs are still ODEs. The training method, initially proposed by Legaard et al. (2023) for NODEs, has been generalized by us to the context of NEVs. This generalization was facilitated by our first contribution, which ensures the well-posedness of the problem and immediately reveals all ODE segments. While other methods could theoretically benefit from our proposed data slicing technique (as Chen et al. (2021) manually performed such slicing in their implementation), we believe our focus on eXplainable AI outweighs performance comparisons with these methods.
>
> 4. This is a legitimate concern, as Legaard et al. (2023) showed many implementations heavily fine-tuned such parameters and cannot be applied as-is in real-world. To debunk any such doubts we perfromed the requested experiments and revised the manuscript.
>
> 5. See 3 above.
>
> **Response to questions:**
> 1. If the reviewer is interested in other automata learning algorithms such as Aalergia, we selected L* because it's a widely recognized active learning algorithm that offers PAC guarantees. If the question pertains to the specific use of L* in this context, we recommend referring to the revised introduction, which explains how the work of Bloem et al. (2020) influenced our choice.
>
> 2. This isn't entirely accurate, particularly when compared to NNs learning the same system. Studies have shown that L* requires up to three orders of magnitude fewer samples in some instances, and it can still enhance NN training (Meinke et al., 2011; Aichernig et al., 2019).
> Furthermore, Automata Learning has been and continues to be the preferred approach for numerous academic and industrial applications, ranging from reverse engineering legacy systems to formally verifying the safety and security of systems. For more information, refer to Vaandrager (2017) and Fisman (2018).
>
> [0] Meinke, K., et al. "Learning-Based Software Testing: A Tutorial." In: ISoLA 2011. Communications in Computer and Information Science. https://doi.org/10.1007/978-3-642-34781-8_16
>
> 3. Such scenarios could be considered SDEs. For SDEs, we can use stochastic automata learning. Since our method is predicated on a change detector, we would first recommend a robust predicate evaluation and only resort to complex automata formalisms if the issue persists. Lastly, it's important to remember that none of the other methods can handle SDEs as they represent a different class of systems, that is outside our scope.
>
> 4. Regrettably, the state-of-the-art is unable to learn a model that is comparably explainable or interpretable. Also, some studies provide artefacts that are tailored to their setup & do not easily extend to other examples.
>
> Regards,\
> Authors

---

### Official Review · Reviewer_tB87 · 2023-11-01

**Soundness:** 2 fair
**Presentation:** 2 fair
**Contribution:** 2 fair
**Rating:** 5
**Confidence:** 2

**Summary:**

This paper proposes an automata learning-based approach to learn piece-wise ODE functions. It defines a set of predicates and builds an abstract model of the system by descritizing the time steps. It detects the discontinuity in the dynamics based on a predicate change detector, slices the trajectories, build the abstract model. Then it learns an abstrct event model and construct an event specification. To learn the piecewise Neural ODE, the paper uses the learned event specification to train a nonlinear transformation to map the last state in prefix to the initial state in the suffix at the discontiuity time.

**Strengths:**

* `Originality`: This paper is original in that it learns piecewise Neural ODE by utilizing a learned event specification.

* `Quality`: There is no technical issue.

**Weaknesses:**

* `Weakness 1`: The proposed approach seems to hinge on whether the human designer has a decent knowledge of the dynamics of the plant so that the designer can provide the predicate for the automata learner to capture the event change.

* `Weakness 2`: The author did not illustrate the specification construction approach in organized manner. All the procedures are described in the pattern of "what I did' rather than explaining the motivation first. There is no soundness and completeness analysis for the automata learning approach in the paper. Let alone the query complexity and efficiency. If the proposed approach can be reduced from the L* algorithm, author should at least illustrate and prove it.

* `Weakness 3`: There is no numerical experiments for the proposed approach. There is no qualitative nor quantitative evaluation for the proposed approach.

**Questions:**

Please address my concerns in the `Weakness` field.

---

> ### Author Response · Authors · 2023-11-23
>
> Dear Reviewer tB87,
>
> Thank you for your time and effort in reviewing our paper. We appreciate your feedback and would like to address your comments and questions.
>
> **On predicates:**
> Thanks for your comment. Our method doesn't strictly require user-provided predicates. The automata learning process only needs the ODE variables. We can write a linear predicate in one variable as $ax + b > 0$, where $a$ and $b$ are symbolic values. This allows us to enumerate symbolic predicates. We iteratively learn an abstract model of the ODE (i.e., $\mathcal{A}_f$), and if $\mathcal{A}_f$ detects an event, we update $\Theta$ and continue with the next symbolic predicate. Later, we learn the actual parameters of the symbolic predicates as described in Section 3.2.
>
> **On Specification construction:**
> You're right. We aimed to use the L* algorithm in its original form. The L* learner differentiates the locations of $\mathcal{E}_f$ based on the immediate output of the function and the state update set returned by the look-ahead mechanism. Transitions that have an empty set of state updates do not specify jumps in the ODE’s dynamics. The motivation for constructing the specification $\mathcal{S}_f$ is to obtain a compressed automaton structure that resembles Hybrid Automata. This enhances the readability and interpretability of the automaton.
> We revised this section in the paper.
>
> **On soundness and completeness:**
> The conflicting behavior of the system as discussed in (Bloem et al., 2020) suggests that if there is a conflicting history then the original L* algorithm is incapable of learning a transducer. In our context, this is not the case as the (arithmetic) functions would not produce any conflicting history so the soundness and completeness of the proposed approach depends on
>
> 1. The L* algorithm that requires the complexity analysis for the Membership and Equivalence queries
> 2. The correctness of the abstraction layer
>
> Regarding the latter, in Section 2.2. we describe the correctness of the abstraction layer.
> Concerning the former, the complexity of the Membership queries and coverage of the Equivalence queries depend on the ODE's characteristics and a variety of factors (e.g., the initial value, and boundary conditions). We are well-aware that we introduced a methodology whose complexity is parameterized by its SUL, though the unconventional nature of this SUL provides an immense opportunity for further studies and future investigations in this line of research to distinguish different complexity classes that might be attributed to distinct dynamic classes. For example, the MQ's complexity depends on the ODE's output vector length and the completeness of the method depends on the coverage on the ODE's state space achieved by the Equivalence Oracle. However, we believe providing such rudimentary speculations about the method's complexity, soundness and completeness will not provide any valuable insight given the current page limits of the submission; therefore, we incorporated this comment in the revision but we will thoroughly address the soundness and completeness concerns in an extended version.
>
> **On experiments:**
> We added two more examples to the paper. We believe the proposed event specification learning also complements other Neural ODE learning methods; therefore, all methods could theoretically benefit from the proposed data slicing. We defer this comment to future work as we think it is essential to focus on proposed theory, and demonstrate its applicability and well-posedness using a simple training. Lastly, some of the state-of-the-art methods do not provide a reproducible, reusable artefact.
>
> Best regards,\
> Authors

---

### Official Review · Reviewer_B45E · 2023-11-01

**Soundness:** 2 fair
**Presentation:** 3 good
**Contribution:** 2 fair
**Rating:** 3
**Confidence:** 3

**Summary:**

The authors propose an extension of the famous L* algorithm for learning deterministic finite automata. The authors adapt this learning algorithm to learn automata for neural event ODEs, which is novel and interesting. A toy example of a bouncing ball is used as a running example to illustrate the approach.

**Strengths:**

The use of L* (or grammatical inference more broadly) to learn automata in the context of neural ODEs is an interest domain.

**Weaknesses:**

The paper misses a description of the actual learning algorithm. It seems to be a variant of L*, but too many details are left to the reader to decipher. The use of L* in the context has also been explored, in particular as part of the reinforcement learning literature, and should be cited. For example, see the references below. As it stands the paper is not sufficiently self-contained or understandable to be recommended for publication. The paper would also greatly benefit from additional experiments.

[1] Dohmen, Taylor, et al. "Inferring Probabilistic Reward Machines from Non-Markovian Reward Signals for Reinforcement Learning." Proceedings of the International Conference on Automated Planning and Scheduling. Vol. 32. 2022.

[2] Topper, Noah, et al. "Active Grammatical Inference for Non-Markovian Planning." Proceedings of the International Conference on Automated Planning and Scheduling. Vol. 32. 2022.

[3] Xu, Zhe, et al. "Joint inference of reward machines and policies for reinforcement learning." Proceedings of the International Conference on Automated Planning and Scheduling. Vol. 30. 2020.

**Questions:**

How is L* being adapted for your setting? I assume it is not a straightforward application of L*. A lot of the formalism provides a foundation for L*, but it is difficult to see how L* has a novel adaptation in this domain.

Minor comments:
* "between the them" -> "between them"
* At the bottom of page 2, you need a space between "define" and "\mathcal{L}(M)}".

---

> ### Author Response · Authors · 2023-11-15
>
> Dear Reviewer B45E,
>
> Thank you for your time and effort in reviewing our paper. We appreciate your feedback and would like to address your comments and questions. Your comments indicate a deep understanding of automata learning, and we believe that this may be a case of misunderstanding on your part.
>
> Firstly, we'd like to clarify that our work does not propose an extension of the L* algorithm for learning DFAs or Mealy machines. Our methodology involves using L* to learn an abstract Mealy machine, with an abstraction layer that is comprehensively described in the paper (please see 2.1.1. and 2.2.2.). Additionally, we incorporate a CacheTree in our Membership Oracle, which employs a simple look-ahead mechanism. This mechanism is designed to discover state updates of the ordinary differential equations (ODEs). The output to each membership query is the valuation of the predicate on the immediate output of the function, plus a set comprising outputs for all extensions of the current query. Such a look-ahead mechanism does not involve extending the L* algorithm niether for DFAs nor for Mealy machines.
>
> **Regarding your questions:**
> 1. In our research, we did not adapt the L* algorithm. Instead, we applied it in the context of abstract automata learning. The application of L* in our setting is straightforward and does not involve any modifications to the algorithm itself.
> 2. Our work does not aim to adapt L* in a novel way. Instead, we leverage the existing properties of L*. As Bloem et al. (2020) demonstrated, in the absence of conflicting history of computations, the L* algorithm converges to a transducer. If there is a conflicting history, the algorithm must split. However, in our case, since (arithmetic) functions produce one value for each input, there should not be any conflicting history in the function's history of computation. If there were, it would not be a function. Therefore, the only adaptation necessary for using L* in this field of research is devising the appropriate abstraction layer.
>
> We acknowledge the importance of the works you cited and agree that they provide a valuable context for our research. We are currently exploring the possibility of extending our work to Stochastic ODEs (SDEs) or Piecewise ODEs with stochastic events, and these references will undoubtedly be beneficial in that endeavor.
>
> In addition to the papers you mentioned, we would also like to draw attention to the following work:
>
> [0] Martin Tappler, et al. "L*-based learning of Markov decision processes (extended version)." Formal Aspects Comput. 33(4-5): 575-615 (2021)
>
> This paper is also particularly relevant to our research and will be included in our citations.
> We will revise the future work section of our paper accordingly and ensure that all relevant works are properly cited.
>
> We’ve addressed the minor comments and fixed the typos in the paper’s PDF.
>
> **We believe there may have been misunderstandings, and we hope our clarifications rectified them. We will also add more clarifications to the paper to avoid such misunderstandings. We look forward to further discussions and your continued feedback.**
>
> Best regards,\
> Authors

---

### Official Review · Reviewer_zGTB · 2023-11-01

**Soundness:** 4 excellent
**Presentation:** 4 excellent
**Contribution:** 3 good
**Rating:** 6
**Confidence:** 4

**Summary:**

The paper introduces a method for learning the behavior of continuous systems with discrete events using abstract automata learning and Neural Event ODEs. It enables the extraction of event specifications and state update functions from black-box ODE implementations, significantly improving training efficiency. This approach aims to reduce training time and computational resources while maintaining the advantages of piecewise solutions for systems with discontinuities.

**Strengths:**

- The paper presents an innovative approach to learning the behavior of continuous systems with discrete events, addressing a challenging problem in the field of modeling and simulation.
- By using abstract automata learning, the method not only captures event specifications but also makes the resulting models interpretable. This is important for understanding complex systems, even when their behavior is initially unknown or represented as black-box ODEs.
- The paper introduces a more efficient training process for Neural Event ODEs by removing discontinuous training pairs. This improvement can lead to reduced training time and computational resource requirements, making it practical for real-world applications.
- The use of the bouncing ball example illustrates the method's effectiveness in simplifying complex systems into interpretable models, making the paper more accessible to a broad audience.

**Weaknesses:**

- The paper mentions preliminary empirical results, but it may lack a comprehensive evaluation of the method's performance on a diverse set of real-world problems. While the paper highlights the relevance of the proposed method for real-world scenarios, it could benefit from concrete examples or case studies demonstrating its application in practical settings.
- The paper might not thoroughly discuss the assumptions and limitations of the proposed method, which is important for understanding its scope and potential constraints.

**Questions:**

More discussions on the bottleneck of the proposed method would further strengthen the paper.

---

> ### Author Response · Authors · 2023-11-23
>
> Dear Reviewer zGTB,
>
> Thank you for your time and effort in reviewing our paper. We appreciate your feedback and would like to address your comments and question.
>
> **Regarding weaknesses**
> 1. We have incorporated two more examples in the paper; namely, a switching dynamical system and a thermostat-controlled storage tank water heater.
>
> 2. We have discussed the weaknesses, applications, limitations and rooms for improvements to the best of our knowledge.\
>         2.1. We made minimal assumptions on the ODE and its events. Subsequently, we used minimal domain or expert information in designing and training the Neural Networks.\
>         2.2. We made no assumptions defining the proposed abstraction layer for automata learning.
>
> **Response the question:**
> Our approach combines the strengths and weaknesses of both Neural Event ODEs and the L* algorithm. The research is in its early stages and requires much attention from the scientific community. Among the limitations, the scalability of the L* algorithm with respect to the size of the input alphabet and the state-space of the system under learning could pose a challenge. Nonetheless, we anticipate that real-world piecewise ODEs will not pose a significant challenge for state-of-the-art automata learning. These methods have demonstrated scalability up to hundreds of input predicates and continuous dynamics (i.e., dynamic locations). For example see https://learnlib.de/performance/libalf-performance-comparison/.
>
> Best regards,\
> Authors

---

### Meta-Review · Area_Chair_YPXQ · 2023-12-24

**Metareview:**

### Summary
This paper presents a hybrid comprehensive approach for inferring an interpretable specification of a system showing piecewise dynamics.
The method uses automata learning to infer an abstract model of a possibly black-box system’s behavior and a neural network to learn its continuous dynamics.
 It enables the extraction of event specifications and state update functions from black-box ODE implementations, significantly improving training efficiency. This approach aims to reduce training time and computational resources while maintaining the advantages of piecewise solutions for systems with discontinuities.

###  Strengths
+ Revisits the problem of learning SLDS with explainable and learning based components using Neural Hybrid Automata.
+ Focus on Learning Representations and Explainable AI (XAI) for piecewise dynamical systems.

### Weaknesses:
- Needs more empirical evaluation beyond the pedagogical bouncing ball example
- Missing comparisons with related works in SLDS with NNs (Poli et al 2021)
- Needs to comment if the approach will scale in representation spaces, since many environments do not have observations in the same variables as the governing SLDS.

**Justification For Why Not Higher Score:**

Reviewers agreed on retaining the scores after discussion, and the AC agrees on limited contributions.

**Justification For Why Not Lower Score:**

All Reviewers are in agreement of the need for such a line of work and appreciate the framework.

---

### Decision · Program_Chairs · 2024-01-16

Reject